# Hyperkalemia in Diabetes Mellitus Setting

**DOI:** 10.3390/diseases10020020

**Published:** 2022-03-28

**Authors:** Kleber Goia-Nishide, Lucas Coregliano-Ring, Érika Bevilaqua Rangel

**Affiliations:** 1Department of Medicine, Nephrology Division, Federal University of São Paulo, São Paulo 04038-901, Brazil; kleber.nishide@unifesp.br (K.G.-N.); lucas.ring@unifesp.br (L.C.-R.); 2Jewish Institute of Research and Education, Albert Einstein Hospital, São Paulo 05652-900, Brazil

**Keywords:** diabetes mellitus, hyperkalemia, kidney and heart disease

## Abstract

Diabetes mellitus is a global health problem that affects 9.3% of the worldwide population and is associated with a series of comorbidities such as heart failure (HF) and chronic kidney disease (CKD). Diabetic patients, especially those with associated CKD, are more susceptible to present potassium disorders, in particular hyperkalemia due to kidney disease progression or use of renin-angiotensin-aldosterone blockers. Hyperkalemia is a potentially life-threatening condition that increases the risk of cardiac arrhythmia episodes and sudden death, making the management of potassium levels a challenge to reduce the mortality rate in this population. This review aims to briefly present the potassium physiology and discuss the main conditions that lead to hyperkalemia in diabetic individuals, the main signs, symptoms, and exams for the diagnosis of hyperkalemia, and the steps that should be followed to manage patients with this potentially life-threatening condition.

## 1. Introduction

Diabetes mellitus (DM) is a globally pandemic metabolic disease with a prevalence of 9.3% worldwide, corresponding to 463 million adults aged 20–79 years in 2019 and with an increasing prevalence to 10.9% of the world population (~700 million) by 2045 [1]. In 2019, it is estimated that 19.3% of people aged 65–99 years (135.6 million) live with diabetes and that number is projected to reach 195.2 million by 2030 and 276.2 million by 2045 [2]. DM is associated with both microvascular complications (kidney disease, retinopathy, and neuropathy) and macrovascular complications (cardiovascular disease, stroke, and peripheral artery disease). It is the leading cause of blindness and amputation and contributes substantially to kidney disease, cardiomyopathy, and cerebrovascular and peripheral artery diseases [3]. Diabetic kidney disease (DKD) is a microvascular complication of DM and the most common cause of end-stage kidney disease (ESKD) worldwide, with approximately 30% of patients with type 1 DM (T1DM) and approximately 40% of patients with type 2 DM (T2DM) developing DKD [4]. Importantly, DKD accounts for cardiovascular complications and the high mortality rate of patients with DM [5].

DM is identified as an independent predictor of hyperkalemia. Many mechanisms contribute to a higher risk of developing hyperkalemia in a DM setting, including impaired potassium (K^+^) excretion, impaired renal tubular function, and a reduced ability to shift K^+^ into cells. In line with these findings, additional risk factors for hyperkalemia encompass advanced chronic kidney disease (CKD), cardiovascular disease, age, and use of renin-angiotensin-aldosterone system (RAAS) inhibitors, which are frequently present in diabetic individuals [6].

In a large population-based cohort of hospitalized HF patients followed up for a medium of 2.2 years, serum K^+^ values > 5.0 mEq/L were found in 39% of patients and were predicted mainly by CKD, DM, and MRA use [7]. To note, these patients were at increased risk of hospitalization (6-month HR, 2.75) and death (HR, 3.39) in comparison with HF patients with normokalemia. Similar findings were documented in patients admitted with acute myocardial infarction and in-hospital mortality (all-cause and composite ventricular fibrillation and cardiac arrest) in accordance with post-admission K^+^ levels [8]. Therefore, K^+^ levels > 5.0 mEq/L or greater were associated with these adverse outcomes, although K^+^ levels lower than 3.5 mEq/L were equally life-threatening.

## 2. Physiology of Potassium (K^+^)

### 2.1. K^+^ Flow between Intracellular and Extracellular Compartments

The ion K^+^ in our body has a key role in maintaining the concentration of charges between the intra- and extracellular environments and in regulating the triggering of action potentials (especially in muscle and nerve cells). Most of the K^+^ in the organism (140–150 mEq/L) is present in the intracellular fluid (ICF), freely dissolved in the cytosol of interstitial and tissue cells. Only a small fraction (2%, e.g., approximately 4 mEq/L) of the K^+^ is found in the extracellular fluid (ECF). The ECF fraction, however, is essential for keeping the body homeostasis and for that reason is particularly regulated by diverse mechanisms. Increasing or decreasing the plasmatic levels of K^+^ (hyperkalemia and hypokalemia, respectively) can lead to severe medical conditions and may require immediate interventions, which we will discuss further ahead in this article.

The total body K^+^ concentration is a result of an equilibrium between the inflow and excretion of the ion (Figure 1). Potassium intake is determined mainly through absorption in the gastrointestinal tract via food ingestion and fluid intake. The main route of K^+^ excretion is through urine, although a small portion is excreted through feces and sweat. The amount of K^+^ ingested per day by an individual is on average 70 mmol. Of this, around 10 mmol is excreted through feces and sweat and 60 mmol is excreted through urine. In the event of vomiting or diarrhea, larger amounts of K^+^ may be lost through the gastrointestinal tract and if hypovolemia develops activation of the RAAS contributes to hypokalemia. The total K^+^ count of ICF is approximately 3000 mmol in comparison to nearly 65 mmol in ECF. Due to this discrepancy in values, a small release of around 1% of the K^+^ from the ICF to ECF would increase the serum concentration of this ion by almost 50% [9].

Likewise, a K^+^-rich meal could easily double or even triple the concentration of K^+^ in the blood would this K^+^ not be quickly transported into the cells [9]. The concentration of K^+^ in the ICF is mainly maintained by the action of the Na^+^-K^+^-ATPase (sodium-potassium ATPase) pump. That pump actively transports three sodium ions to the extracellular medium and two potassium ions into the cells. The skeletal muscle is the tissue that holds the greatest amount of K^+^ in the body due to the large volume of fluid that this tissue stores. The musculature has a great capacity of buffering the K^+^ present in the blood, either capturing or releasing this ion into the extracellular environment. Two hormones that actively participate in this process are insulin and adrenaline, which, through stimulation and increase of expression of the Na^+^-K^+^-ATPase pump in the cell membranes, promote K^+^ uptake by tissues. The gastrointestinal tract also influences K^+^ balance through the stimulation of complex neuronal networks, which regulate the release of incretins and other hormones by enteroendocrine cells and mediate the communication with the central nervous system, influencing the transport of K^+^ in various targets including the kidneys [9].

### 2.2. Regulators of K^+^ Transport and Levels

After a meal, the rapid release of insulin from the pancreas promotes the entry of K^+^ that has just been absorbed from the gastrointestinal tract into cells. Slowly, this amount of K^+^ that has been stored in the cells is released into the bloodstream, from where it will be filtered by the kidneys and excreted in the urine. In addition to the blood glucose concentration, an increase in the plasma K^+^ concentration is by itself a stimulus for the release of insulin by the pancreatic ß cells. Additionally, the insulin-promoted transport of glucose into cells is an energy source that supports the functioning of the Na^+^-K^+^-ATPase pump, maintaining the K^+^ gradient between the intra- and extracellular environments [9].

Adrenaline, in turn, has its physiological importance on K^+^ transport during periods of stress or physical activity. It is precisely at these times that K^+^ is being released out of muscle and nerve cells, which are collectively triggering action potentials. Strenuous exercise, for example, can transiently increase blood K^+^ concentrations for that reason. The release of adrenaline, especially by the adrenal glands, is a counterregulatory mechanism to this phenomenon. Adrenaline increases the activity of Na^+^-K^+^-ATPase in peripheral tissues to normalize the K^+^ concentrations. In a scenario of tissue damage and cell death, K^+^ can be released from within the cells into the blood by the loss of barrier function that was exerted by the plasma membrane. This situation is counteracted by the release of adrenaline in response to trauma and stress, which will promote the transport of K^+^ into healthy tissues [9].

Another mechanism that regulates K^+^ concentrations between the intra- and extracellular compartments is the blood hydrogen (H^+^) concentration. An increase in the H^+^ in the blood induces the release of K^+^ from the cells, while a decrease in the H^+^ of the blood leads to an entry of K^+^ into the cells. For every 0.1 unit decrease in blood pH there is a 0.6 mEq/L increase in serum K^+^ concentration. On the other hand, each 0.1 unit increase in pH is accompanied by a 1 mEq/L decrease in serum K^+^ concentration. This asymmetric effect is due to differences in fluid volume between the intra- and extracellular environments and due to the different concentrations of K^+^ in each of these locations. A protein that plays an important role in the balance of H^+^ and K^+^ in the body is the hydrogen-potassium-ATPase (H^+^-K^+^-ATPase) pump, located in the parietal cells of the gastric mucosa and the apical membrane of intercalated cells of the collecting duct. This pump promotes K^+^ reabsorption and H^+^ release under the influence of K^+^ balance, aldosterone, and acid-base balance [9].

### 2.3. The Potassium Transport System in the Kidneys

About 90% of the K^+^ that has been filtered into Bowman’s space is reabsorbed by the proximal convoluted tubule and by the thick ascending segment of Henle’s loop. In situations of hyperkalemia, in which the body’s physiological response is to eliminate K^+^, a large amount of the ion is excreted in the distal segments of the nephron, leading to a large concentration of K^+^ in the urine (Figure 2) [9].

An individual with a glomerular filtration rate (GFR) of 100 mL/min/1.73 m^2^ is capable of filtering between 145 and 150 L of plasma per day. Under these conditions, the total amount of K^+^ filtered over a day by the kidneys is around 600 mEq/L if the serum K^+^ concentration is at physiological levels of 4 mEq/L [9].

The main mechanism of K^+^ reabsorption in the proximal segment of the nephron involves the paracellular route after a concentration gradient has been established with the reabsorption of Na^+^ and water, which concentrates the remaining solutes in the tubular lumen. This system, therefore, is regulated by the amount of Na^+^ and water absorbed from the tubular lumen. In the proximal tubule cells, the Na^+^-K^+^-ATPase pump present in the basolateral membrane promotes Na^+^ reabsorption into the interstitium at the expense of their exchange for K^+^. These K^+^ ions, which were concentrated in the cytosol of tubular cells, are returned into the interstitium by facilitated diffusion through K^+^ channels also present in the basolateral membrane [9].

Potassium reabsorption continues in the thick ascending limb (TAL) of Henle’s loop. Tubular cells in this segment express multicarrier channels of sodium-potassium-chloride (Na^+^-K^+^-2Cl^−^) in their apical membrane. These channels are capable of carrying one sodium, one potassium, and two chlorides ions (two positive and two negative charges), keeping the charge ratio across the membrane unchanged. Renal outer medullary channel (ROMK) channels are ATP-dependent potassium channels located across the apical and luminal membranes of tubular cells. These channels play an important role in K^+^ recycling in the TAL of Henle’s loop and K^+^ secretion at the cortical collecting duct (CCD) of the nephron [10].

In the thick segment, there is also the transport of K^+^ from the interstitium to the cytosol of the tubular cells through the Na^+^-K^+^-ATPase pumps. Finally, the presence of K^+^-Cl^−^ symport channels in the basolateral membrane of tubular cells in the thick segment of Henle’s loop captures part of the K^+^ infused by the Na^+^-K^+^-ATPase pump and returns it to the interstitial environment. Together, these mechanisms are responsible for reabsorbing an additional 25% of the K^+^ filtered in the glomerulus; these, alongside 65% that has already been proximally reabsorbed, result in only around 10% of the filtered K^+^ remaining to be reabsorbed in the distal segments of the nephron [9].

The distal nephron consists of many segments; the most important for the transport and excretion of K^+^ being the convoluted distal tubule and the collecting duct. These nephron segments have both the ability to reabsorb and to secrete K^+^ so that the effective excretion of K^+^ in the urine is a result of a balance between these two processes. The two most common cell types that are present both in the last segment of the distal tubule and in the collecting duct are the principal cells (~70%) and the intercalated cells [9].

The principal cells are responsible for the secretion of K^+^, performing an uptake of K^+^ from the interstitium into the tubular cells by Na^+^-K^+^-ATPase pumps and subsequent release into the tubular lumen. These cells express ROMK channels, which release K^+^ from the cytosol of these cells into the tubular lumen. Potassium reabsorption is performed by type A intercalated cells, through H^+^-K^+^-ATPase pumps, responsible for the uptake of K^+^ and release of H^+^ in the tubular lumen [11]. The small remaining amount of K^+^ to be reabsorbed is performed in the medullary collecting ducts, maintaining the effective excretion of K^+^ per day between 70 and 75 mEq/L under physiological conditions [9].

Aldosterone acts by promoting Na^+^ reabsorption and K^+^ secretion in the kidney by binding to the mineralocorticoid receptors (ENaC, Na^+^-K^+^-ATPase pumps and ROMK channels) located in collecting ducts, increasing its transcriptional activity [12]. The net effect is that aldosterone functions homeostatically to maintain normal sodium and potassium balance, as well as blood pressure and circulating blood volume [13].

## 3. Hyperkalemia and Diabetes Mellitus

### 3.1. Definition

Hyperkalemia is a state defined by high serum K^+^ levels (above 5.5 mEq/L), which can lead to a series of symptoms and complications. The main causes that lead to this condition are the inability of kidneys to correctly excrete K^+^, high K^+^ intake (rarely the sole cause of hyperkalemia, since healthy individuals can adapt to the excess of K^+^ consumption by increasing renal excretion [14]), and impairment of the cellular flux of K^+^ (inflow and outflow), leading to a shift of K^+^ from the ICF to ECF [15].

### 3.2. Main Causes of Hyperkalemia in Individuals with DM

In comparison with the general population, patients with DM are more susceptible to hyperkalemia due to a series of alterations in the diabetic environment such as hyporeninemic hypoaldosteronism, hyperosmolality, insulin deficiency, and the use of medications to treat comorbidities such as K^+^-sparing drugs [16].

#### 3.2.1. Hyporeninemic Hypoaldosteronism

Hyporeninemic hypoaldosteronism is a syndrome caused by a reduction in the synthesis and secretion of renin by juxtaglomerular cells in the kidneys. This reduction in renin liberation leads to a dysfunction in the RAAS, leading to a reduction in the secretion of aldosterone by the adrenal glands. Aldosterone acts by promoting the reabsorption of Na^+^ and secretion of K^+^ into the lumen of the cortical collecting duct, thereby having a major role in the regulation of K^+^ in the body. The most common risk factor for developing hyporeninemic hypoaldosteronism is DM and there is a series of factors in the diabetic individual that contributes to this reduction in the release of renin in the system [16,17]:Injury to the juxtaglomerular apparatus (responsible for renin synthesis and release) due to DKD;Impaired conversion of prorenin to active renin (molecular mechanisms remain to be elucidated, but are probably correlated to sympathetic dysfunction);Autonomic dysfunctions caused by the autonomic diabetic neuropathy;Chronic renal salt retention, leading to volume expansion, which causes an increase in the release of atrial natriuretic peptide, promoting suppression of renin secretion and inhibition of RAAS.

In most cases, a combination of these mechanisms is most likely responsible for the rise of the hyporeninemic hypoaldosteronism syndrome in diabetic patients, resulting in hyperkalemia [16,17].

#### 3.2.2. Hyperosmolality

Patients with uncontrolled DM also have increased plasma osmolality due to hyperglycemia. The concentration gradient established by this condition promotes an outflux of water from inside the cells to the interstitium, and K^+^, the most abundant intracellular cation, is carried by the water outside of the cells, elevating its serum concentration [14,18].

Some of the main and potentially fatal conditions that may lead to this state of hyperosmolality in diabetic patients are the Hyperglycemic Hyperosmolar State (HHS) and Diabetic Ketoacidosis (DKA) [14,18].

#### 3.2.3. Variations of Insulin and Glucagon Concentrations on Potassium Regulation

Patients with T1DM and some with T2DM have insufficient to insignificant insulin secretion. Insulin is a hormone that promotes K^+^ cellular uptake by a series of mechanisms (such as translocation and activation of Na^+^-K^+^-ATPase and inhibition of K^+^ efflux), playing a major role in the K^+^ homeostasis, especially after exogenous K^+^ loads, by using intracellular buffering to reduce hyperkalemia before renal excretion. Patients with T2DM who have insulin resistance but normal production and secretion of insulin by the pancreas, however, do not present an increased risk of hyperkalemia by these mechanisms, since insulin independently regulates glucose and K^+^ uptake into cells (in the insulin resistance of T2DM only glucose uptake is compromised, whereas K^+^ uptake is preserved) [18].

To note, recent studies suggest that glucagon also plays an important role in the K^+^ regulation alongside insulin, promoting urinary excretion of K^+^ in the distal nephron and the collecting duct. Table 1 shows the combined action of both insulin and glucagon in the serum glucose and K^+^ regulation [19]:

#### 3.2.4. Medications

Diabetic patients usually present other comorbidities alongside diabetes mellitus such as hypertension and eventually make use of medications that can lead to hyperkalemia. The most prominent medication in this category is spironolactone, an aldosterone receptor blocker that reduces urinary excretion of K^+^ by blocking the action of aldosterone, which can then lead to hyperkalemia. The incidence of hyperkalemia induced by spironolactone usage is significantly higher in patients with CKD (relatively common in diabetic patients due to microvascular changes and DKD) and when used in combination with ACEi (angiotensin-converting enzyme inhibitor) or ARB (angiotensin II receptor blocker). In these cases, serum K^+^ and renal function should be frequently monitored [20].

According to KDIGO (Kidney Disease: Improving Global Outcomes), ACEi or ARB are recommended for patients with DM, hypertension, and albuminuria; these medications should be titrated according to patient tolerance [21]. Therefore, blood pressure, serum creatinine, and serum K^+^ should be monitored within 2–4 weeks of initiation or after a dose increase of those drugs. If serum creatinine rises by less than 30% and K^+^ is normal within 4 weeks following initiation of ACEi or ARB treatment, it is possible to continue these drugs on a maximally tolerated dose or to increase the dose accordingly. Conversely, if hyperkalemia is observed it is recommended to review concurrent drugs, recommend moderate K^+^ intake, and consider prescribing diuretics, sodium bicarbonate, and gastrointestinal cation exchangers and reduce the dose or stop ACEi or ARB as a last resort. Additionally, if serum creatinine raises by more than 30%, review potential causes of acute kidney injury, correct volume depletion, reassess concomitant medications (diuretics or nonsteroidal anti-inflammatory drugs), consider renal artery stenosis, and reduce the dose or stop ACEi or ARB as a last resort. Nonetheless, hyperkalemia associated with ACEi or ARB is often managed by measures to reduce serum K^+^ levels rather than decreasing the dose or immediately stopping these drugs. To note, the combination of ACEi and ARB is potentially harmful in the DKD setting as this combination leads to an increase in adverse events, in particular hyperkalemia (6.3 events per 100 person-years vs. 2.6 events per 100 person-years with monotherapy) and acute kidney injury (12.2 vs. 6.7 events per 100 person-years) [22]. Importantly, women who are receiving ACEi or ARB therapy should be advised regarding contraception and discontinuation of these drugs in women who are considering pregnancy or who become pregnant [21].

Mineralocorticoid receptor antagonists, in particular finerenone, are effective in reducing albuminuria in patients with DKD treated with a RAS blocker, while having smaller effects on serum K^+^ levels than spironolactone [23]. In the study “Finerenone in Reducing Kidney Failure and Disease Progression in DKD” (FIDELIO-DKD), finerenone use resulted in a lower risk of CKD progression and cardiovascular events in individuals with T2DM and DKD [24]. Moreover, the incidence of hyperkalemia-related discontinuation of finerenone was higher than with the placebo (2.3% and 0.9%, respectively). In the FIGARO-DKD study, among patients with T2DM treated with RAS blockers and stage 2 to 4 CKD with moderately elevated albuminuria or stage 1 or 2 CKD with severely elevated albuminuria, finerenone therapy improved cardiovascular outcomes (death from cardiovascular causes, nonfatal myocardial infarction, nonfatal stroke, or hospitalization for heart failure), as compared with the placebo [25]. As observed in the FIDELIO-DKD study, the incidence of hyperkalemia-related discontinuation of the trial regimen was higher with finerenone (1.2%) than with the placebo (0.4%).

However, when serum K^+^ levels were analyzed in the FIDELIO-DKD study, over 2.6 years’ median follow-up, 21.4% of patients treated with finerenone experienced treatment-emergent ≥ mild hyperkalemia (defined as K^+^ > 5.5 mEq/L) as opposed to 9.2% in the placebo group [26]. This effect occurred in a time-dependent manner. Independent risk factors for mild hyperkalemia comprised higher serum K^+^, lower eGFR, increased urine albumin-to-creatinine ratio, younger age, female sex, ß-blocker use, and finerenone assignment. Moderate hyperkalemia (defined as K^+^ > 6.0 mEq/L) occurred in 4.5% of the patients in the finerenone group when compared with 1.4% in the placebo group. Importantly, diuretic or sodium-glucose co-transporter-2 (SGLT2) inhibitor use reduced the risk of hyperkalemia. Despite the increase in serum K^+^, at 4 months, after routine K^+^ monitoring and K^+^ management strategies, the impact of increased hyperkalemia risk for any change from baseline was smaller with finerenone when compared with the placebo group [26].

Other common medications that may cause hyperkalemia include: K^+^ supplements, trimethoprim (due to blockage of Na^+^ channel in distal nephron), calcineurin inhibitors (tacrolimus due to inhibition of aldosterone production and cyclosporine due to blocking Na^+^-K^+^-ATPase activity in the distal nephron), heparin (due to inhibition of aldosterone production), digoxin (inhibition of extrarenal K^+^ disposal by blocking Na^+^-K^+^-ATPase activity in skeletal muscle), ß-blockers (inhibition of extrarenal K^+^ disposal by blocking ß2-adrenergic receptors), and nonsteroidal anti-inflammatory agents. Drug-induced hyperkalemia occurs most often in patients with impaired kidney function and associated hyporeninemic hypoaldosteronism and it is especially common in the elderly [27].

#### 3.2.5. Pseudohyperkalemia

It is important to take note of pseudohyperkalemia, a condition marked by a false elevation in measured serum K^+^ levels (>0.4 mmol/L) as compared with the normal plasma potassium concentration, which should always be considered before aggressive treatment, especially in cases where serum K^+^ levels are high without any evident reason or clinical evidence of electrolyte imbalance [28].

Pseudohyperkalemia is a laboratory artifact rather than a biological disorder [29] and occurs due to the release of potassium from cells and platelets during the processes of specimen collection and clot formation. Some of the main reasons for the occurrence of this phenomenon are hemolysis during specimen collection (the use of tourniquet, excessive fist-pumping during blood collection, and the use of a syringe when compared with a vacuum device can all increase the risk of hemolysis during blood draw), contamination by a potassium-containing substance (such as potassium ethylenediaminetetraacetic acid [K2-EDTA]), and specimen transport and clinical conditions such as leukocytosis and thrombocytosis (often associated with false elevation of potassium level) [28,30].

### 3.3. Symptoms, Investigation, and Diagnosis

Hyperkalemia is established when serum K^+^ is equal to or higher than 5.5 mmol/L and can be classified according to its levels into mild (5.5–6.5 mmol/L), moderate (6.5–8 mmol/L), and severe (>8 mmol/L) hyperkalemia [31].

#### 3.3.1. Symptoms

Neuromuscular symptoms are the most common manifestations of hyperkalemia, including muscle weakness, nausea, fasciculations, and paresthesia of superior and inferior members. More severe and acute cases may result in ascending paralysis (with eventual flaccid quadriplegia) and cardiac condition abnormalities, which can lead to dysrhythmias and even death. Most patients with mild to moderate chronic hyperkalemia, however, are asymptomatic. Head, trunk, and respiratory muscles are usually spared and respiratory failure is very rare [30,32].

In physiological conditions, serum K^+^ levels in the normal range (3.5–4.5 mEq/L) may impact axonal membrane function by interfering in nerve excitability, which comprises super-excitability and the responses of depolarization and hyperpolarization currents [33]. In ESKD patients, serum K^+^ of 5.0 mEq/L is associated with excitability abnormalities of nerve membrane which promotes preferentially depolarization, despite the reduction of uremic toxins after dialysis treatment [34]. Importantly, dietary K^+^ restriction in stage 3–4 CKD patients contributed to a reduction in neuropathy severity and progression when pain, numbness, weakness, gait speed, and quality of life scores were evaluated and compared with the group of standard care [35]. These findings shed light on the amelioration of nerve function in ESKD patients when normokalemia is achieved.

From a pathophysiologic perspective, K^+^ can be considered as a uremic neurotoxicity molecule, as reviewed elsewhere [36]. Therefore, nerve and muscle membranes become depolarized as the potassium equilibrium potential (E_k_) is shifted by ~12 mV in the depolarizing direction when serum K^+^ reaches 7.0 mEq/L and ~8 mV when serum K^+^ reaches 6.0 mEq/L in comparison with normal resting level of K^+^ (4.5 mEq/L). When hyperkalemia is present, clinical features are characterized by ascending paralysis, quadriplegia, areflexia, and abnormalities of the cranial nerves [36].

#### 3.3.2. Exams and Diagnosis

The first test that should be evaluated in patients suspected of hyperkalemia is the electrocardiogram (ECG), since cardiac conditions caused by hyperkalemia can be lethal (Figure 3) [37]. Elevated levels of serum K^+^ have several consequences for the myocardial action potential and may manifest as a series of progressive ECG abnormalities such as [17,30,32]:Peaked T waves (tall, narrow, and symmetrical). Sometimes can be confused with hyperacute T wave change associated with ST-segment elevation myocardial infarction; however, the T waves in the latter condition used to be more asymmetric and broad based;ST-segment elevation;Widening of PQ (PR) interval and QRS complex;Loss of P wave;In more severe cases (K^+^ > 8 mEq/L), we can encounter sinusoidal wave patterns, indicative of the development of ventricular fibrillation and asystole.

It is important to note, however, that the plasma K^+^ concentration often correlates poorly with cardiac manifestations, while the rate of rising of serum K^+^ plays a much bigger role in the development of these abnormalities than the concentration level per se. Patients with chronic hyperkalemia may present relatively normal ECGs even at elevated levels of serum K^+^, whereas patients with acute hyperkalemia may have significant ECG alterations at much lower serum K^+^ levels [30,32].

Additional laboratory tests should be directed towards causes suggested by physical examination and history. These additional tests include [15,16,30,31]:Blood urea nitrogen (BUN) and creatinine to assess renal function. Urine K^+^, Na^+^, and osmolality may also help investigate the cause;Patients with renal disease should be tested for serum calcium (Ca^2+^) levels, since hypocalcemia may exacerbate cardiac alterations of hyperkalemia;Complete blood count to screen for hemolytic anemia, leukocytosis, or thrombocytosis;Serum glucose, glycosylated hemoglobin, and blood gas analysis for diabetic patients and those with suspected acidosis;Lactate dehydrogenase for patients with suspected hemolysis;Creatinine phosphokinases and urine myoglobin for patients with suspected rhabdomyolysis;Uric acid and phosphorus for patients with suspected tumor lysis syndrome;Digoxin serum levels for patients on digoxin, since digoxin toxicity may cause hyperkalemia;Cortisol and aldosterone levels if no other cause is found, to assess for mineralocorticoid deficiency. Hyporeninemic hypoaldosteronism should be considered in diabetic patients with hyperkalemia and low serum aldosterone and accounts for most cases of unexplained chronic hyperkalemia in patients in whom GFR and K^+^ intake would not be expected to result in hyperkalemia. To confirm and perform a differential diagnosis, PRA (plasma renin activity), serum aldosterone, and serum cortisol should be measured (hyporeninemic hypoaldosteronism is usually characterized by low PRA and normal serum cortisol);Transtubular Potassium Gradient (TTKG) is a formula used to determine whether hyperkalemia is caused by aldosterone deficiency or resistance or whether hyperkalemia is secondary to nonrenal causes and is expected to be high (usually >10) in hyperkalemia with a normal renal response. An inappropriately low TTKG in a hyperkalemic patient suggests hypoaldosteronism or a renal tubule defect;Pseudohyperkalemia should always be considered and confirmed in asymptomatic patients without typical ECG abnormalities before initiating aggressive therapy.

In Figure 4, we provide a diagnostic algorithm based on GFR and aldosterone levels [31]:

### 3.4. Management and Treatment

The management of hyperkalemia depends primarily on the rate of development of the condition, signals and symptoms associated (especially electrocardiographic changes and neuromuscular symptoms), cause, and the absolute serum K^+^ level [30,32].

If abnormalities in the ECG are present, a cardiac stabilizer such as calcium gluconate should be readily administered before managing the hyperkalemia per se to reduce mortality [38].

The treatment of this condition consists of three main steps that should be done in sequence (cardiac stabilization, promotion of K^+^ shift into cells, and elimination of K^+^ from the body), followed by monitoring of K^+^ and prevention of future recurrence [38].

#### 3.4.1. Cardiac Stabilization

The ratio of K^+^ in ICF to ECF enables the generation of action potentials and is essential for the normal functions of neurons, skeletal muscles, and cardiac muscles. Sudden onset and rapid progression of hyperkalemia (due to decreased excretion, excessive intake or shift of K^+^ from ICF to ECF, or even a combination of these factors) is a life-threatening condition. As K^+^ levels increase in the ECF, the magnitude of the K^+^ gradient across the cell membrane is reduced, as well as the absolute value of the resting membrane potential. Thus, membrane voltage becomes less negative, moving closer to the threshold potential, leading to an easier initiation of the action potential and making myocytes more excitable. However, the further rise of K^+^ has the opposite effect as the value of the membrane potential at the onset of an action potential determines the number of voltage-gated Na^+^ channels activated during depolarization. As the value of membrane potential becomes less negative in hyperkalemia, the number of available Na^+^ channels decreases, resulting in a slower influx of Na^+^ and subsequently slower impulse conduction and decreasing myocytes excitability [17,39,40]. In Figure 5 we illustrate these mechanisms:

Therefore, the first step in managing hyperkalemia when ECG alterations are present or the plasma K^+^ level is equal or higher than 6.0 mEq/L consists in the administration of calcium. Calcium salts act by antagonizing the potassium-induced decrease in membrane excitability of cardiomyocytes, preventing the development of potentially lethal arrhythmias and stabilizing the cardiac response to hyperkalemia; however, these drugs do not alter plasma K^+^ concentration per se [27,30,32,41].

Calcium is usually administered as an intravenous injection of 10 to 20 mL of 10% calcium gluconate over 5–10 min. If ECG changes persist after 5–10 min of the administration, the second injection of calcium should be given after 5 min [41,42]. Calcium chloride 10% may also be used in some situations, as it contains three times more elemental calcium than calcium gluconate 10% (total calcium 270 mg/10 mL or 13.6 mEq/10 mL versus 10 mg/10 mL or 4.65 mEq/10 mL, respectively). Nonetheless, calcium chloride is more irritating to peripheral vessels and may cause tissue necrosis with extravasation, therefore it is only administered through central venous lines or peripherally in cardiac arrest. Due to these reasons, calcium gluconate is usually the preferred choice in patients with signals of cardiac toxicity [30].

It is also important to be aware of patients using digitalis since hypercalcemia can increase the cardiotoxic effects of these drugs. In these cases, 10 mL of 10% calcium gluconate is added to 100 mL of 5% dextrose in water and infused over 20–30 min to avoid hypercalcemia, and these patients should be closely monitored [41].

#### 3.4.2. Promotion of K^+^ Shift into Cells

##### Insulin

Insulin plays an important role in lowering plasma K^+^ concentration by binding to its receptor on skeletal muscle, promoting an increase in the abundance and activity of Na^+^-K^+^-ATPase, thus causing K^+^ to shift into cells. Although patients with T2DM have resistance to the glycemic effect of insulin, its ability to enhance K^+^ uptake by skeletal muscle and liver is not affected [27].

It usually takes 10–20 min for insulin to start lowering serum K^+^ levels, with maximal action in 60 min, and the effect lasts for 2–6 h. The most common regimen is 10 units of regular insulin accompanied by a 25–50 g infusion of glucose as intravenous injection, increased to 60 g if 20 units of insulin are used. Hyperglycemic patients (serum glucose > 300 mg/dL) can be given insulin alone to avoid worsening hyperkalemia caused by the hyperosmolar state [30,32,41]. Glycemic levels should always be closely monitored in all patients to avoid hypoglycemia; furthermore, recent studies have shown that, because of the increased risk of developing hypoglycemia, doses of insulin administered in patients with CKD or ESKD should be lowered to 5 units to avoid this complication [43].

##### Beta-2 Agonists

Beta-2 receptor agonists, such as albuterol or salbutamol, produce an effect similar to insulin, shifting K^+^ into cells by increasing Na^+^-K^+^-ATPase pump activity, especially in skeletal muscle cells [32]. It can be administered by inhalation, nebulization, or intravenously and the dose administered by inhalation is 4 to 8 times those prescribed for bronchodilation in the treatment of acute asthma. High doses of albuterol may stimulate both α-receptors (which cause K^+^ release from the liver and may temporarily increase serum K^+^ by >0.4 mmol/L) and ß-1 receptors (which can lead to arrhythmias). Subcutaneous terbutaline may also be used to lower serum K^+^ in mildly hyperkalemic patients with ESKD [27].

Beta-2 agonists may also be used in combination with insulin to promote a significantly greater fall in K^+^ levels than when either agent is individually administered and hypoglycemia is less likely to occur with the combined therapy than when insulin alone is used [27].

##### Sodium Bicarbonate

Sodium bicarbonate is only recommended in patients with metabolic acidosis and can be administered in a bolus dose of 50 mEq/L, or as continuous effusion [41], or even as a dose of 1–2 mEq/Kg or 1–2 mL/kg (for sodium bicarbonate 8.4% solution). Its administration promotes the uptake of K^+^ by skeletal muscle by favoring sodium-bicarbonate cotransport and sodium-hydrogen exchange, which increases intracellular Na^+^, and thus promotes an increase in the activity of Na^+^-K^+^-ATPase, shifting K^+^ into cells [27].

To note, in individuals with CKD the efficacy of sodium bicarbonate to ameliorate hyperkalemia is dubious as it takes at least 3 to 4 h for the serum K^+^ to start to decrease in dialysis patients, as opposed to immediate action in patients with normal renal function. In addition, sodium bicarbonate can also cause volume overload, which is of paramount importance in CKD setting, and its administration does not enhance the potassium-lowering effects of insulin and ß2-agonists [44,45].

#### 3.4.3. Promotion of K^+^ Elimination from the Body

Strategies to promote K^+^ shift into cells are only temporary maneuvers and should be followed by strategies to eliminate excess K^+^ from the body itself [27].

First of all, ambulatory patients with chronic hyperkalemia should be oriented to see a nutritionist to guide food choices. It is also important to investigate whether the patient is taking any K^+^ supplements or medications that limit K^+^ excretion, since temporary dose reduction or discontinuation of these medications may be all that is needed to reestablish normokalemia. Patients with more severe hyperkalemia, however, require active measures to eliminate K^+^ from the body. Some of these measures are described below [27]:

##### Urinary K^+^ Excretion

Diuretic drugs such as loop or thiazide diuretics can be used in patients with moderately compromised kidney function to substantially enhance urinary K^+^ excretion by increasing flow and delivery of Na^+^ to the collecting duct. Thiazide diuretics are most effective when the eGFR is greater than 30 mL/min/1.73 m2, whereas loop diuretics should be administered in patients with more severe renal insufficiency. These medications have been used to manage extremely severe hyperkalemia without dialysis; however, diuretics only work if the patients have adequate kidney function [27,30,32].

Furosemide can be administered at a dose of 1–1.5 mg/kg (1 mg/kg for loop diuretic naïve patients and 1.5 mg/kg in patients previously exposed to loop diuretics) as it starts to act in 15–30 min and has a half-life of 4–6 h. Therefore, in early acute kidney injury settings, the furosemide stress test indicates a novel assessment of tubular function with a robust predictive capacity to identify those patients with severe and progressive acute kidney injury [46].

Although Na^+^ retention and the potential for adverse mineralocorticoid effects on the myocardium make chronic fludrocortisone unattractive for long-term therapy, it can be useful acutely as it begins to take effect within 3 h from administration. Large doses of fludrocortisone (up to 0.4 mg daily) may be required as patients with kidney disease may have aldosterone resistance as well as aldosterone deficiency [27].

In patients with normal renal function, SGLT2 inhibitors are known to enhance potassium excretion by the kidney through a combination of mechanisms (increased sodium and water delivery to the distal nephron, enhanced glycosuria, and stimulation of aldosterone) and are also known to provide cardiorenal protection in patients with CKD. Among patients with T2DM and CKD treated with RAAS blockade, recent studies have shown that SGLT2 inhibition with canagliflozin may also reduce the risk of hyperkalemia without increasing the risk of hypokalemia [47].

##### Dialysis

Hemodialysis is the most effective way to eliminate the excess of K^+^ from the body and is the therapy of choice for life-threatening hyperkalemia in patients with ESKD or severe renal impairment, severe rhabdomyolysis, or severe hyperkalemia that is not responsive to medical management [27,30,41].

Plasma K^+^ usually drops by 1 mmol/L during the first hour of treatment, with a total drop of about 2 mmol/L in 3 h, and then reaches a nadir, remaining stable at 4 h. Replenishment from cellular stores continues when K^+^ removal stops, therefore there is a substantial post-dialysis rebound of plasma K^+^, proportional to the pre-dialysis K^+^ level. Electrolytes should be carefully monitored for at least 24 h after hemodialysis administration [27,41].

##### Sodium Polystyrene Sulfonate

Sodium polystyrene sulfonate (SPS) is a cation exchange resin that acts primarily in the large intestine by exchanging sodium for K^+^ ions (along with Ca^2+^, ammonium, and magnesium [Mg^2+^]) [27]. SPS is usually administered by retention enema or orally in combination with sorbitol to avoid bowel obstruction and to speed its delivery to the distal colon where it is most effective [27,41].

The resin, however, is not used as a first-line treatment for hyperkalemia because of its slow onset of action (>2–4 h) and lack of immediate effect. Long term use is not recommended since SPS is poorly tolerated and has been linked to a series of complications such as intestinal ischemia and colonic necrosis with increased morbimortality, especially in elderly patients [27,30,32,41].

##### Patiromer

Patiromer is a non-absorbable synthetic polymer that, unlike SPS, does not swell significantly when exposed to water and does not require a laxative such as sorbitol to reach the distal colon where it is most effective. This drug binds to K^+^ (as well as ammonium and Mg^2+^) in exchange for Ca^2+^ in the gastrointestinal tract, especially in the colon, lowering plasma K^+^ concentration in a dose-dependent manner by increasing fecal K^+^ loss (15 to 30 g/day administered orally have been shown to increase daily fecal K^+^ by 15 to 20 mmol) [27,32]. The onset of action is around 7 h and doses range from 8.4 to 25.2 g per day.

The main side effects caused by the drug are constipation and hypomagnesemia, which can be readily treated with magnesium supplementation, but overall the drug is well tolerated by most patients. Theoretically, because patiromer exchanges Ca^2+^ for K^+^ in the distal colon, it has the potential of causing positive Ca^2+^ balance and ectopic calcifications, however longer-term studies are necessary to disprove this concern [27,32].

##### Sodium Zirconium Cyclosilicate

Sodium zirconium cyclosilicate (ZS-9) is an investigational drug highly selective for K^+^ and ammonium ions through mechanisms very similar to those occurring in ion channels. It exchanges Na^+^ and H^+^ ions for K^+^, binding the latter throughout the gastrointestinal tract and increasing fecal K^+^ losses in a dose-dependent manner. Adverse events were usually comparable to those with placebo in clinical trials, but in addition to minor gastrointestinal side effects edema developed in some patients, especially when higher doses were used [27,32].

Even though there are some concerns about hypomagnesemia and positive Ca^2+^ balance from patiromer and Na^+^ overload from ZS-9, both agents are effective and well-tolerated when taken chronically [27].

## 4. Conclusions

In this article, we presented the main reasons why diabetic patients are so vulnerable to developing hyperkalemia, the mechanisms behind it, and the current methods of treatment and management of this potentially severe condition. Rapid identification of K^+^ disorders can prevent the occurrence of serious life-threatening complications such as cardiac arrhythmias and respiratory muscle impairment.

The treatment of this condition consists of three main steps that should be done in sequence (cardiac stabilization, promotion of K^+^ shift into cells, and elimination of K^+^ from the body by using diuretics, exchange resins, and/or dialysis), followed by monitoring of K^+^ and prevention of future recurrence.

## Figures and Tables

**Figure 1 diseases-10-00020-f001:**
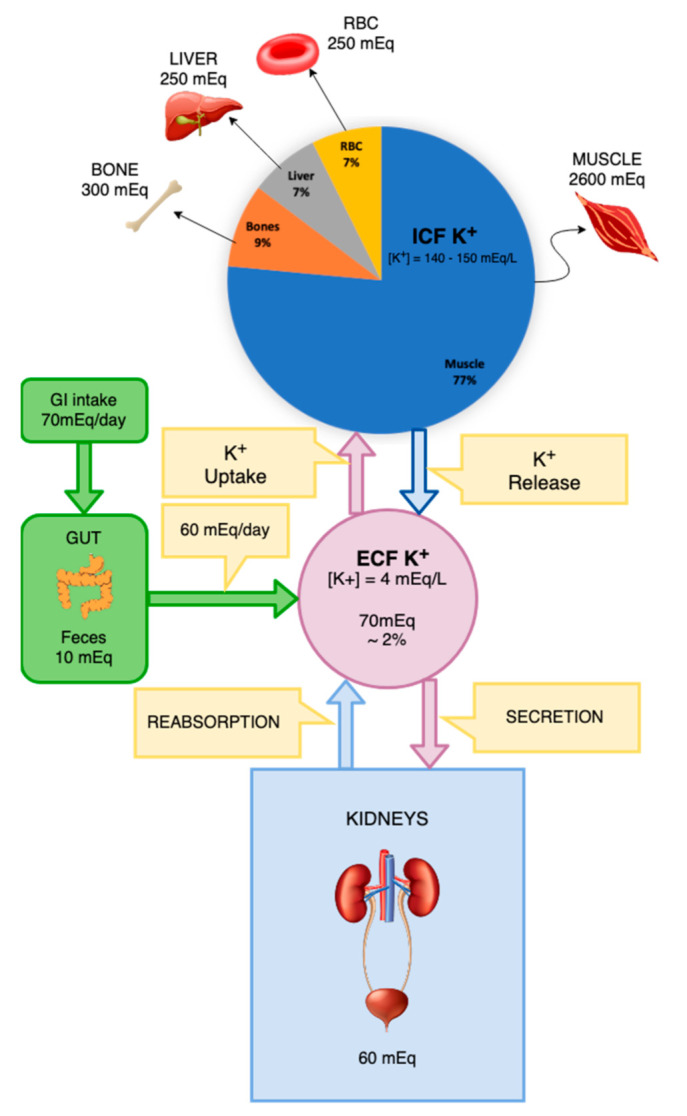
Distribution of potassium in different body compartments and its inflow and secretion routes.

**Figure 2 diseases-10-00020-f002:**
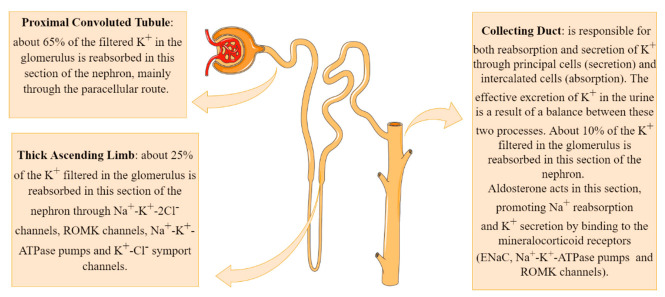
Potassium transport system in the kidneys.

**Figure 3 diseases-10-00020-f003:**
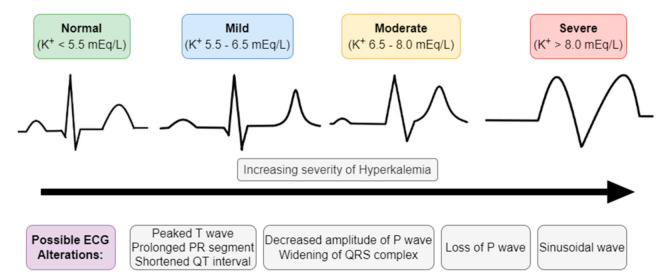
Electrocardiographic (ECG) manifestations of hyperkalemia.

**Figure 4 diseases-10-00020-f004:**
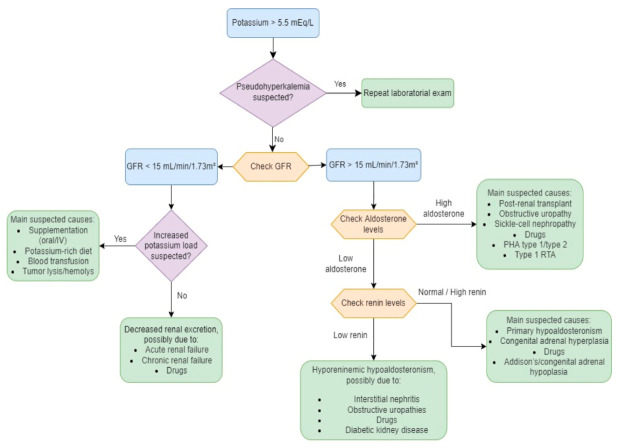
Diagnostic algorithm in hyperkalemia. Adapted and modified from [31]. GFR—glomerular filtration rate; RTA—renal tubular acidosis; PHA—pseudohypoaldosteronism.

**Figure 5 diseases-10-00020-f005:**
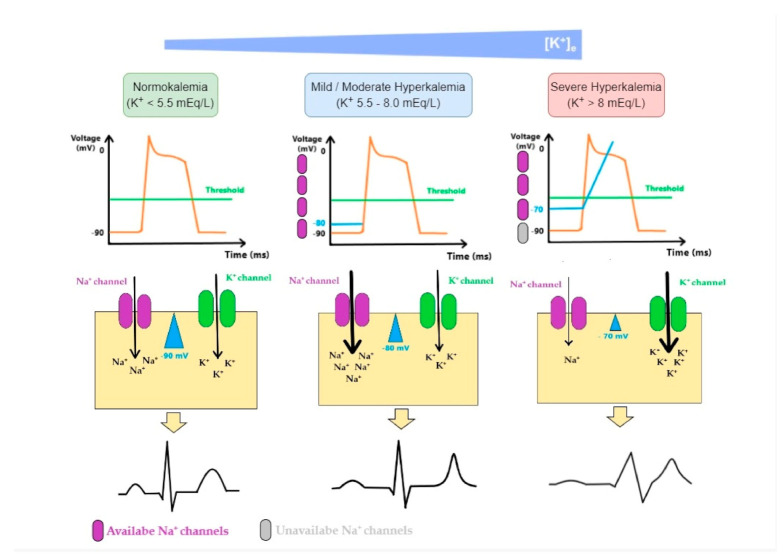
Mechanisms of cardiac arrhythmia in hyperkalemia: As K^+^ levels increase in the ECF (extracellular fluid), the magnitude of K^+^ gradient across the cell membrane is reduced and so is the absolute value of the membrane potential. Membrane potential becomes less negative moving closer to the threshold potential, making it easier to initiate an action potential. The effect it has on the excitability of myocytes depends on K^+^ levels. While initial changes seem to increase cardiomyocyte excitability, a further rise in K^+^ has the opposite effect. This is because the value of the membrane potential at the onset of an action potential determines the number of fast voltage-gated Na^+^ channels activated during the depolarization. As this value becomes less negative in hyperkalemia, the number of available fast Na^+^ channels decreases, resulting in a lower influx of Na^+^ and, subsequently, slower impulse conduction. These changes are associated with an increase in K^+^ equilibrium potential (E_K_). ECG changes produced by hyperkalemia follow a typical pattern that generally correlates with K^+^ serum levels: when fast Na^+^ channels are activated, an increase in excitability and conduction velocity is observed in ventricular cardiomyocytes and early repolarization occurs synchronously, which leads to a peaked-T wave; as K^+^ increases in ECF, the inactivation of fast voltage-dependent Na^+^ channels and the activation of K^+^ channels lead to reductions in conduction velocity and can render cells refractory to excitation, which promotes P wave widening and flattening, PR interval widening, QRS complex widening and eventually blending with T wave, generating, therefore, a sinus-wave; therefore, these changes comprise repolarization defects, conduction delays, paralysis of atria, and junctional/ventricular rhythms.

**Table 1 diseases-10-00020-t001:** Combined effects of insulin and glucagon on potassium and glucose regulation (intermediate situations between “high” and “low” levels of each hormone are also possible). Adapted and modified from [19].

Condition	Insulin	Glucagon	Glucose Regulation	Potassium Regulation
Post prandial state (several hours after a meal)	Low	Low	Modest gluconeogenesis, providing glucose for basal metabolism	No effect
Fast	Low	High	Gluconeogenesis from endogenous AAs for sustaining glucose needs of the body	Potassium excretion issued from the cells from which AAs were catabolized for gluconeogenesis
Carbohydrate rich meal	High	Low	Metabolism and/or storage of the ingested glucose	No effect
Meat meal (protein and potassium rich), potassium load or potassium rich meal	High	High	Increased gluconeogenesis from ingested AAs *. Metabolism and/or storage of the newly formed glucose (from meat meals)	Insulin-dependent storage of potassium inside cells, followed by progressive release resulting from glucagon-induced increase in urinary potassium excretion

* AAs: Amino acids.

## Data Availability

Data sharing is not applicable to this article as no new data were created or analyzed in this study.

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
