# Peer review of "Hyperkalemia in Diabetes Mellitus Setting"

_diseases, 2022, doi:10.3390/diseases10020020_

Round 1
Reviewer 1 Report
The review titled: Hyperkalemia in diabetes mellitus setting is a good work and illustrated most established papers about hyperkalemia and diabetes, how to diagnose and and report the case.
In section 2.1.....Authors discussed the K+ flow between intra-cellular and extracellular compartments, i suggest you to add section titled regulators of K transport and levels after section 2.1. then continue with 2.2 about K+ transport in kidney.
Section 3.2.5 pseudhyperkalemia need more information and is short.
Of note, number of references for the current review are few compared to it contents ............increase and support most sentence with recent references.
Figure are hard to follow try to simplify them especially figure 3.
If possible try to summarize your explained sections with more figures and or tables.
Author Response
We would like to thank the reviewers for their comments, which certainly contributed to strengthening our manuscript. We reviewed the manuscript accordingly and highlighted the changes using the Track Changes tool throughout the manuscript.
The review titled: Hyperkalemia in diabetes mellitus setting is a good work and illustrated most established papers about hyperkalemia and diabetes, how to diagnose and and report the case.
1) In section 2.1.....Authors discussed the K+ flow between intra-cellular and extracellular compartments, i suggest you to add section titled regulators of K transport and levels after section 2.1. then continue with 2.2 about K+ transport in kidney.
Response: As suggested, in page 3, 2nd paragraph, we added the subsection “2.2 Regulators of K+ transport and levels”.
2) Section 3.2.5 pseudhyperkalemia need more information and is short.
Response: Thank you for the suggestion. We expanded this topic in page 9 (3rd and 4th paragraph) in section “3.2.5 Pseudohyperkalemia”.
3) Of note, number of references for the current review are few compared to it contents ............increase and support most sentence with recent references.
Response: We revised the paper accordingly and added more references and supported most sentences and/or paragraphs with them.
4) Figure are hard to follow try to simplify them especially figure 3.
Response: Thank you for pointing this out. We have revised figure 3, as suggested (now figure 4).
5) If possible try to summarize your explained sections with more figures and or tables.
Response: Thank you very much for the suggestion. We have added/modified 4 figures in the article (figures 2, 3, 4 and 5), which include potassium transport system in the kidneys, ECG abnormalities, diagnostic algorithm in hyperkalemia, and mechanisms of cardiac arrhythmia.
Sincerely,
Érika B Rangel, MD, PhD
Reviewer 2 Report
The manuscript titled “Hyperkalemia in diabetes mellitus setting” discussed the relationship between hyperkalemia and DM in detail and the symptoms, diagnosis, and management of this disease. This article may improve the recognition of this disease. The following suggestions should be considered to improve the manuscript’s overall quality and readability.
1. There are too many segmentations, although the authors described the same thing be forcibly divided into several paragraphs—for example, the first three paragraphs in the Introduction section. The 2-4th and the 5-6th paragraphs in 3.2.4 should be combined into one paragraph. Too many same issues that are difficult to list here.
2. Please use the superscript word to exhibit the positive and negative ions, such as “K+.”
3. “2. Physiology of Potassium”:
1) “2.2. The potassium transport system in the kidneys”: It’s better to add a figure for good understanding.
2) The authors are recommended to add the sub-sections for “The potassium transport system in the cardiac muscles” and “The potassium transport system in the neurons.”
4. Table 1 shows the normal homeostasis of glucagon and insulin on potassium regulation and doesn’t belong to the “3.2.3. Insulin Deficiency” section.
5. Among the total 4 Figures and Tables, Table 1 and Figure 2, 3 are duplicated the cited literature with no or minor modifications. The authors are recommended to look for more references and summarize by themselves.
Author Response
We would like to thank the reviewers for their comments, which certainly contributed to strengthening our manuscript. We reviewed the manuscript accordingly and highlighted the changes using the Track Changes tool throughout the manuscript.
The manuscript titled “Hyperkalemia in diabetes mellitus setting” discussed the relationship between hyperkalemia and DM in detail and the symptoms, diagnosis, and management of this disease. This article may improve the recognition of this disease. The following suggestions should be considered to improve the manuscript’s overall quality and readability.
- There are too many segmentations, although the authors described the same thing be forcibly divided into several paragraphs—for example, the first three paragraphs in the Introduction section. The 2-4thand the 5-6thparagraphs in 3.2.4 should be combined into one paragraph. Too many same issues that are difficult to list here.
Response: Thank you for the suggestion. We revised the text accordingly and combined the indicated paragraphs.
- Please use the superscript word to exhibit the positive and negative ions, such as “K+.”
Response: We revised the text and used the superscript word to exhibit all the positive and negative ions, as suggested.
- “2. Physiology of Potassium”:
1) “2.2. The potassium transport system in the kidneys”: It’s better to add a figure for good understanding.
Response: Thanks for the suggestion. We added a novel figure (figure 2) to summarize the section. Furthermore, we added 2 new paragraphs to the text, one in the introduction (page 2, 1st paragraph), and one to briefly explain aldosterone action (page 5, 4th paragraph).
2) The authors are recommended to add the sub-sections for “The potassium transport system in the cardiac muscles” and “The potassium transport system in the neurons.”
Response: Thank you for bringing these topics to discussion. In page 10, we added a new paragraph to discuss the effect of potassium on neural cells (2nd paragraph).
In the section “3.4.1 .Cardiac Stabilization”, we added a figure (Figure 5) to better explain the potassium transport system within the cardiac muscles.
- Table 1 shows the normal homeostasis of glucagon and insulin on potassium regulation and doesn’t belong to the “3.2.3. Insulin Deficiency” section.
Response: We changed the title of the section to “Variations of insulin and glucagon concentrations on potassium regulation”
- Among the total 4 Figures and Tables, Table 1 and Figure 2, 3 are duplicated the cited literature with no or minor modifications. The authors are recommended to look for more references and summarize by themselves.
Response: Thank you for the suggestion. We have revised figures 2 and 3 (now 3 and 4, respectively), and added 2 new figures to the articles (figures 2 and 5).
Sincerely,
Érika B Rangel, MD, PhD
Round 2
Reviewer 1 Report
All comment have been answered and well explained.
Reviewer 2 Report
The authors have addressed my concerns well, and now this manuscript can be accepted for publication.